# Association between estimated pulse wave velocity and stroke in middle-aged and elderly Chinese: A nationwide cohort study

Xiaomei Zheng[1,2], Yixian Zhang[1], Pengsheng Chen[3], Shengren Xiong [4]*

1 Department of Rehabilitation medicine, Fujian Medical University Union Hospital, Fuzhou, China,
2 Fujian Medical University, Fuzhou, China, 3 Department of Plastic Surgery and Regenerative Medicine, Fujian Medical University Union Hospital, Fujian Medical University, Fuzhou, China, 4 Department of Orthopaedics, Fujian Medical University Union Hospital, Fujian Medical University, Fuzhou, China

* Xsrenzj@163.com

## Abstract

### Background

Research on the relationship between estimated pulse wave velocity (ePWV) and stroke risk remains limited. Therefore, this study aimed to investigate the association between ePWV and the risk of stroke in middle-aged and older adults in China.

### Methods

This cohort study included 9,190 participants from the China Health and Retirement Longitudinal Study (CHARLS) in 2011, with follow-up assessments in 2013, 2015, and 2018. ePWV values were categorized into tertiles. Stroke was defined as the occurrence of stroke during follow-up. Longitudinal associations between ePWV and stroke risk were evaluated using Cox proportional hazards regression models.

### Results

In the fully adjusted model, compared to the first tertile, both the second and third tertiles were significantly associated with an increased risk of stroke, with a 40% higher risk for the second tertile (HR = 1.40, 95% CI: 1.17–1.76, P = 0.004) and a 42% higher risk for the third tertile (HR = 1.42, 95% CI: 1.08–1.86, P = 0.012). Restrictive cubic spline analysis further demonstrated a nonlinear dose-response relationship between ePWV and stroke risk (P nonlinear = 0.001).

### Conclusions

Our findings suggest that elevated ePWV is significantly associated with an increased risk of stroke in middle-aged and elderly Chinese individuals. ePWV can

**Data availability statement:** All relevant data arewithin the paper and its Supporting Information files.

**Funding:** The author(s) received no specific funding for this work.

**Competing interests:** The authors have declared that no competing interests exist.

screen individuals at high risk of stroke early in clinical practice and guide personalized prevention and intervention strategies.

## 1 Introduction

Stroke is one of the leading causes of death and long-term disability worldwide, posing a significant challenge to public health [1]. In China, stroke is the top contributor to disability-adjusted life years (DALYs), with approximately 3.94 million new cases and 2.19 million stroke-related deaths annually [2,3]. Given the high prevalence of stroke and its profound socioeconomic impact, early identification of stroke risk factors is critical for developing effective prevention strategies [4,5].

Arterial stiffness, which reflects the elasticity and functional state of blood vessels, is commonly assessed by measuring the speed of pulse wave propagation through the arterial system [6–8]. Currently, carotid-femoral pulse wave velocity (cfPWV) is widely regarded as the "gold standard" for evaluating arterial stiffness [6,9]. However, measuring cfPWV requires specialized equipment and trained personnel, which limits its application in large-scale clinical practice [9]. To overcome this limitation, the Arterial Stiffness Collaboration developed a mathematical model based on age and blood pressure to estimate pulse wave velocity (ePWV), providing a noninvasive method for assessing aortic stiffness [10]. Studies have shown that ePWV predicts major cardiovascular events with accuracy comparable to cfPWV [11].

Existing literature has established that ePWV is independently associated with cardiovascular events and all-cause mortality [12,13]. However, research on the relationship between ePWV and stroke risk remains limited. Most existing studies are focused on western populations [12,14], while fewer studies have been conducted in Asian populations (especially Chinese populations). Therefore, this study aimed to investigate the association between ePWV and the risk of stroke in middle-aged and older adults in China, using data from the nationally representative China Health and Retirement Longitudinal Study (CHARLS). The findings provide new insights and scientific evidence for the early prevention and intervention of stroke.

## 2 Methods

### 2.1 Study population and design

The study participants were middle-aged and elderly adults from CHARLS, an ongoing, nationally representative, population-based, prospective, longitudinal study (http://charls.pku.edu.cn/). The primary aim of this study was to recruit participants aged 45 years and above, but some participants aged 40–44 years also attended the baseline survey. Details of the study design have already been reported [15]. CHARLS employed a multistage, stratified, probability sampling method, covering 450 communities in 150 districts across 28 provinces. Baseline data collection was completed between June 2011 and March 2012, enrolling 17,708 participants, with follow-ups conducted biennially. Data were collected using standardized questionnaires. The study was approved by the Biomedical Ethics Review Committee of Peking University (IRB00001052–11015), and all participants provided written informed consent.

This study analyzed data from the baseline survey and the 2013, 2015, and 2018 follow-up waves. Among the 17,708 baseline participants, we excluded individuals who were under 45 years of age or had missing age information (n = 777), lacked ePWV data (n = 3,689), lacked stroke data (n = 46), had a history of stroke or cardiac events at baseline (n = 411), or were lost to follow-up (n = 3,595). The final analysis included 9,190 participants. The detailed screening process is illustrated in Fig 1.

## 2.2 Assessment of ePWV

ePWV was calculated using the formula proposed by Greve et al., derived from reference values established by the Arterial Stiffness Collaboration [10,11].

$$ePWV = 9.587 - (0.402 \times age) + [4.560 \times 0.001 \times (age2)] - [2.621 \times 0.00001 \times (age2) \times MBP] + (3.176 \times 0.001 \times age \times MBP) - (1.832 \times 0.01 \times MBP)$$

Blood pressure was measured three times using an electronic sphygmomanometer (Omron HEM-7200) with participants in a seated, relaxed position, and the average of the three readings was used. Participants were categorized into tertiles of ePWV: low (T1), medium (T2), and high (T3).

## 2.3 Ascertainment of stroke

The primary outcome of interest was incident stroke, defined as the occurrence of stroke during follow-up. Stroke events were identified through self-reported responses to the standardized question: "Has a doctor ever told you that you have been diagnosed with a stroke?" A "yes" response was considered indicative of an incident stroke [16,17].

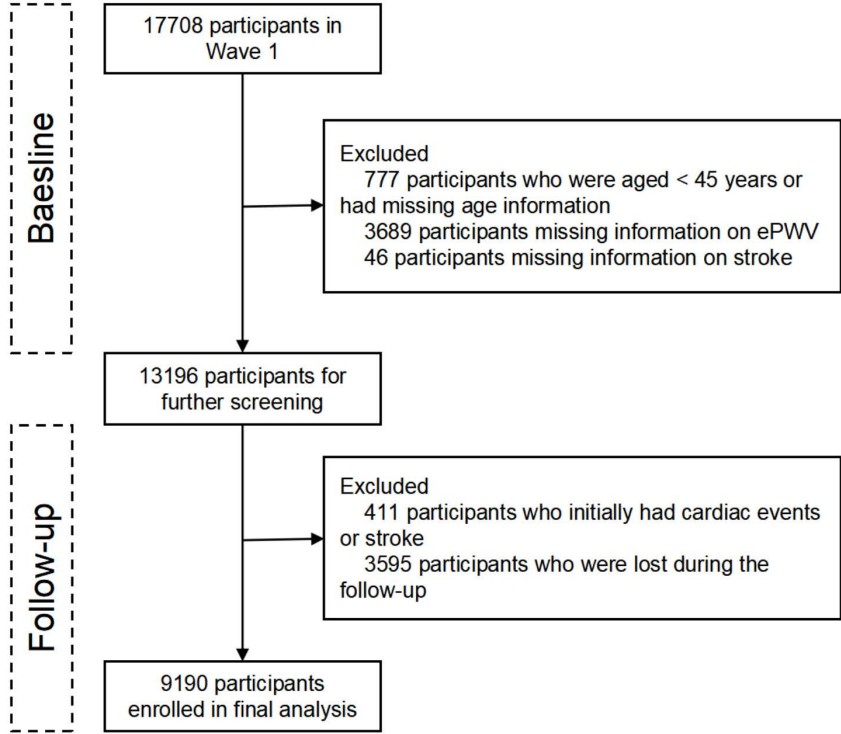

**Fig 1. Flowchart of study participant inclusion.**

## 2.4 Covariates

The covariates included age, sex (male/female), residence (urban/rural), educational attainment (illiterate, primary, secondary, high school, or above), and marital status (married/other). Additionally, lifestyle and metabolic factors were considered, including smoking status (yes/no), alcohol consumption (yes/no), body mass index (BMI), hypertension, diabetes, and dyslipidemia. Smoking, alcohol consumption, and BMI were selected as potential confounders in the relationship between ePWV and stroke risk based on previous studies [18,19]. BMI was classified according to the World Health Organizatio criteria: underweight (<18.5 kg/m²), normal weight (18.5–24.9 kg/m²), overweight (25.0–29.9 kg/m²), and obese (≥30.0 kg/m²). Hypertension was defined as a physician-diagnosed condition, systolic blood pressure ≥140 mmHg, diastolic blood pressure ≥90 mmHg, or the use of antihypertensive medication [20]. Diabetes was defined as a physician diagnosis, fasting plasma glucose ≥126 mg/dL, hemoglobin A1c ≥ 6.5%, or the use of antidiabetic medication [21]. Dyslipidemia was defined as a physician-diagnosed condition, total cholesterol level ≥240 mg/dL, or the use of lipid-lowering medication [22].

## 2.5 Statistical analysis

Continuous variables were presented as means ± standard deviations (SDs), and categorical variables were expressed as frequencies and percentages. Baseline characteristics were compared across ePWV tertiles using chi-square tests or Fisher's exact tests for categorical variables and analysis of variance (ANOVA) for continuous variables. Cox proportional hazards regression models were employed to assess the association between ePWV and incident stroke risk, estimating hazard ratio (HR) and 95% confidence interval (CI). Model 1 was unadjusted. Model 2 accounted for demographic and socioeconomic factors, including age, sex, residence, marital status, and educational attainment. Model 3 was further adjusted for lifestyle and cardiometabolic risk factors, including smoking status, alcohol consumption, BMI, hypertension, diabetes, and dyslipidemia, to minimize potential confounding. Restricted cubic spline (RCS) analysis was conducted to explore the dose-response relationship between ePWV and stroke risk (three knots at the 10th, 50th, and 90th percentiles of ePWV). Statistical analyses were conducted using R software (version 4.2.2) and Zstats v1.0 (www.zstats.net). A two-sided p-value of <0.05 was considered statistically significant.

## 3 Results

A total of 9,190 participants were included in the analysis, with a mean (SD) age of 58.28 (8.75) years. Of these, 4,227 (46%) were male, and 4,963 (54%) were female. During the follow-up, 678 participants (7.38%) experienced a stroke. Participants in the lowest ePWV tertile were more likely to be non-smokers and non-drinkers, have a normal BMI, and be free of hypertension, diabetes, dyslipidemia and stroke (P < 0.05). The baseline characteristics of participants, stratified by ePWV tertiles, are summarized in Table 1.

The cumulative incidence of stroke among participants with diffrent ePWV group was depicted using Kaplan-Meier curves (Fig 2). The Log-rank test revealed a significant difference in stroke incidence rates among the tertiles (P < 0.001). Cox proportional hazards regression analysis showed a significant association between ePWV and the risk of incident stroke. In the unadjusted model (Model 1), Compared to the first tertile, both the second and third tertiles were significantly associated with an increased risk of stroke, with a 86% higher risk for the second tertile (HR = 1.86, 95%CI: 1.50–2.31, P < 0.001) and a 148% higher risk for the third tertile (HR = 2.48, 95%CI: 2.02–3.04, P < 0.001). In the fully adjusted model (Model 3), Compared to the first tertile, both the second and third tertiles were significantly associated with an increased risk of stroke, with a 40% higher risk for the second tertile (HR = 1.40, 95% CI: 1.17–1.76, P = 0.004) and a 42% higher risk for the third tertile (HR = 1.42, 95% CI: 1.08–1.86, P = 0.012) (Table 2).

Model 3 adjusted for Model 2 + smoking status, alcoholing status, body mass index, hypertension, diabetes and dyslipidemia

**Table 1. Characteristics of study participants according to ePWV.**

| Variables | Total | ePWV | | | P value |
|---|---|---|---|---|---|
| | | T1 | T2 | T3 | |
| No. of participants | 9190 | 3064 | 3063 | 3063 | |
| Age, Mean±SD | 58.28±8.75 | 51.82±5.12 | 57.02±6.03 | 66.00±8.01 | <0.001 |
| Sex, n(%) | | | | | <0.001 |
| Male | 4227 (46.00) | 1248 (40.73) | 1431 (46.72) | 1548 (50.54) | |
| Female | 4963 (54.00) | 1816 (59.27) | 1632 (53.28) | 1515 (49.46) | |
| Residence, n(%) | | | | | 0.458 |
| Rural | 6141 (66.82) | 2066 (67.43) | 2054 (67.06) | 2021 (65.98) | |
| Urban | 3049 (33.18) | 998 (32.57) | 1009 (32.94) | 1042 (34.02) | |
| Education Level, n(%) | | | | | <0.001 |
| Illiteracy | 4358 (47.42) | 1246 (40.67) | 1425 (46.52) | 1687 (55.08) | |
| Primary school | 2032 (22.11) | 615 (20.07) | 670 (21.87) | 747 (24.39) | |
| Middle school | 1865 (20.29) | 804 (26.24) | 653 (21.32) | 408 (13.32) | |
| High school or above | 935 (10.17) | 399 (13.02) | 315 (10.28) | 221 (7.22) | |
| Marital status, n(%) | | | | | <0.001 |
| Married | 991 (10.78) | 158 (5.16) | 229 (7.48) | 604 (19.72) | |
| Others | 8199 (89.22) | 2906 (94.84) | 2834 (92.52) | 2459 (80.28) | |
| Smoking status, n(%) | | | | | <0.001 |
| No | 5643 (61.40) | 1999 (65.24) | 1874 (61.18) | 1770 (57.79) | |
| Yes | 3547 (38.60) | 1065 (34.76) | 1189 (38.82) | 1293 (42.21) | |
| Alcoholing status, n(%) | | | | | <0.001 |
| No | 5657 (61.59) | 1975 (64.52) | 1872 (61.12) | 1810 (59.13) | |
| Yes | 3528 (38.41) | 1086 (35.48) | 1191 (38.88) | 1251 (40.87) | |
| BMI, n(%) | | | | | <0.001 |
| Underweight | 581 (6.46) | 171 (5.69) | 156 (5.20) | 254 (8.51) | |
| Normal | 5610 (62.37) | 2001 (66.57) | 1809 (60.26) | 1800 (60.28) | |
| Overweight | 2346 (26.08) | 726 (24.15) | 866 (28.85) | 754 (25.25) | |
| Obesity | 457 (5.08) | 108 (3.59) | 171 (5.70) | 178 (5.96) | |
| Hypertension, n(%) | 3610 (39.28) | 549 (17.92) | 1194 (38.98) | 1867 (60.95) | <0.001 |
| Diabetes, n(%) | 1121 (12.21) | 301 (9.83) | 408 (13.35) | 412 (13.46) | <0.001 |
| Dyslipidemia, n(%) | 1572 (17.18) | 415 (13.61) | 566 (18.60) | 591 (19.33) | <0.001 |
| Stroke, n(%) | 678 (7.38) | 128 (4.18) | 237 (7.74) | 313 (10.22) | <0.001 |

ePWV: Estimated pulse wave velocity, SD: standard deviation

Restrictive cubic spline analysis further demonstrated a nonlinear dose-response relationship between ePWV and stroke risk (P nonlinear=0.001), suggesting that elevated ePWV may significantly increase the risk of incident stroke in middle-aged and elderly populations (Fig 3).

## 4 Discussion

This study utilized nationwide cohort data from the CHARLS to systematically explore the association between ePWV and stroke risk in middle-aged and elderly individuals. Our findings indicate that higher ePWV is an independent predictor of increased stroke risk. The dose–response analysis showed a nonlinear association of ePWV and the hazard of stroke.

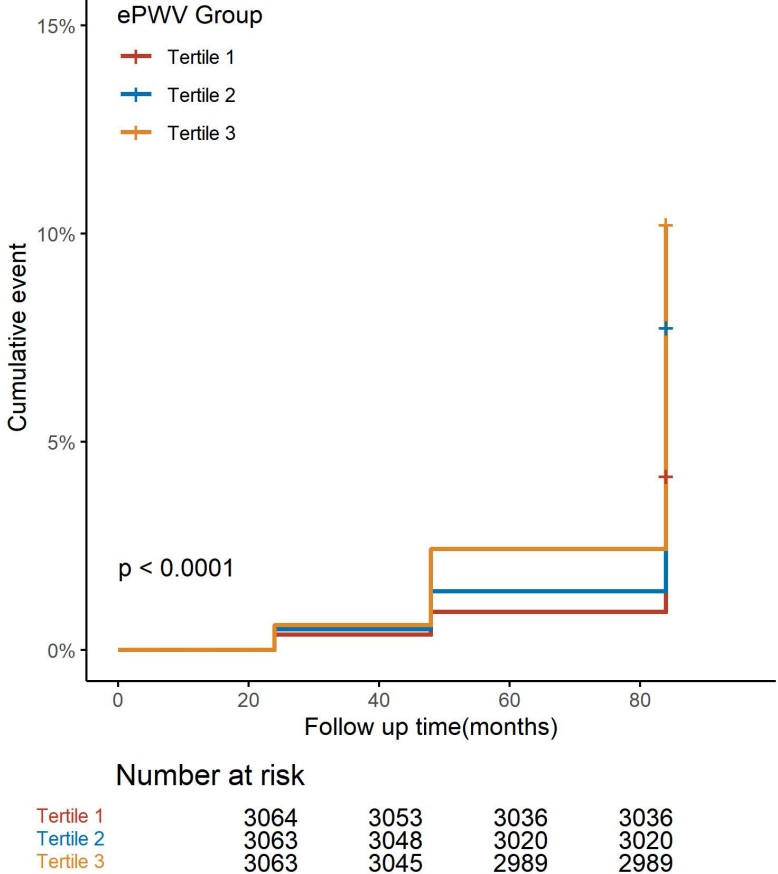

**Fig 2. Kaplan–Meier curves illustrating the cumulative risk of stroke by ePWV.** ePWV: Estimated pulse wave velocity.

**Table 2. Associations of ePWV with the risk of stroke.**

| Variables | Model 1 | | Model 2 | | Model 3 | |
|---|---|---|---|---|---|---|
| | HR (95%CI) | P value | HR (95%CI) | P value | HR (95%CI) | P value |
| ePWV | | | | | | |
| T1 | 1.00 (Reference) | | 1.00 (Reference) | | 1.00 (Reference) | |
| T2 | 1.86 (1.50–2.31) | <0.001 | 1.77 (1.42–2.21) | <0.001 | 1.40 (1.11–1.76) | 0.04 |
| T3 | 2.48 (2.02–3.04) | <0.001 | 2.16 (1.66–2.80) | <0.001 | 1.42 (1.08–1.86) | 0.012 |

ePWV: Estimated pulse wave velocity, HR: Hazard Ratio, CI: Confidence Interval.

Model 1 was unadjuasted

Model 2 adjusted for Model 1+age, gender, residence, marital status, and educational level

Previous studies have shown a positive correlation between aortic stiffness (assessed noninvasively via cfPWV) and stroke risk [23,24]. This study further extends this field by highlighting ePWV as a simple and convenient surrogate marker of arterial stiffness, demonstrating its association with stroke risk. This finding aligns with several prior studies. For instance, a Finnish cohort study on middle-aged men reported that ePWV was associated not only with overall stroke risk but also with ischemic and hemorrhagic stroke subtypes [14]. Similarly, research by Xu et al. supported the association

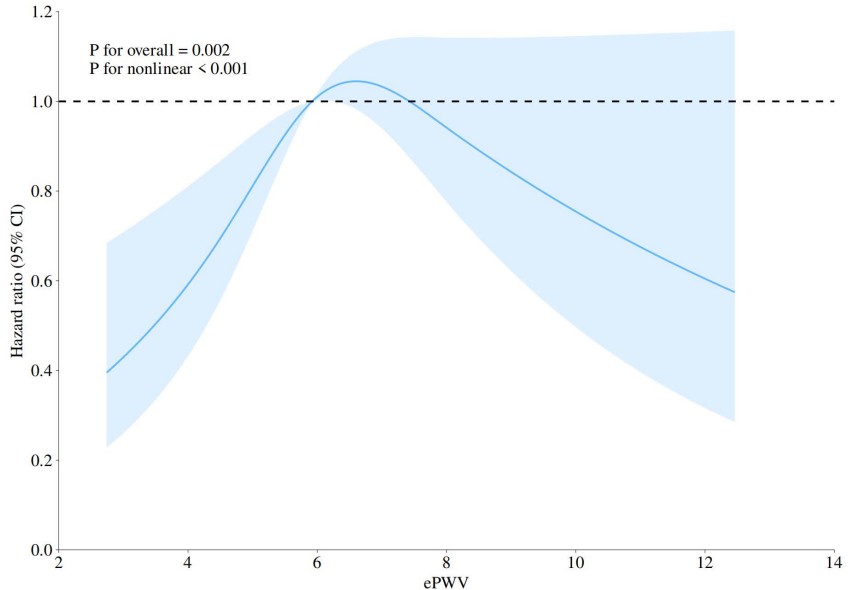

**Fig 3. Nonlinear dose-response relationship between ePWV and stroke incidence.** This Figure presents hazard ratios (HRs) for stroke incidence based on Cox proportional hazards regression model 3, adjusted for age, sex, residence, marital status, educational attainment, smoking, alcohol consumption, body mass index, hypertension, diabetes, and dyslipidemia. The solid line represents the HR, and the shaded area indicates the 95% confidence interval (CI).

between elevated ePWV and increased risk of both stroke and its subtypes [25]. These findings reinforce the link between arterial stiffness and stroke, suggesting that ePWV could serve as a potential tool for early identification of individuals at high risk for stroke, providing valuable insights for clinical management and prevention.

The formation of arterial stiffness is a key feature of atherosclerosis, influenced by various factors such as changes in vascular wall structure and elevated blood pressure [26,27]. As individuals age, the loss of arterial elasticity increases the risk of arterial stiffening. Additionally, chronic hypertension further exacerbates arterial stiffness [26]. In our study, participants with higher ePWV exhibited significant differences in smoking, and alcohol consumption, and cardiovascular metabolic factors. Unhealthy lifestyles such as smoking and drinking increase oxidative stress and systemic inflammation, which accelerate the process of arterial atherosclerosis [28,29]. Future research should investigate the potential biological mechanisms underlying the association between elevated ePWV and stroke, including endothelial dysfunction and the formation of atherosclerotic plaques.

This study has several strengths. First, it is based on a large sample cohort of middle-aged and elderly individuals in China, enhancing the external validity and generalizability of the results. Second, the prospective cohort design with long-term follow-up enables reliable assessment of stroke incidence. Third, this study fills a gap in the literature regarding the relationship between ePWV and stroke risk, offering new insights for early identification and prevention of stroke in middle-aged and elderly populations. Furthermore, the study emphasizes the importance of monitoring ePWV in clinical practice, which could help optimize stroke risk management strategies.

However, there are some limitations. First, the generalizability of our findings may be limited by the lack of racial and regional diversity in the study sample. As the CHARLS cohort consists exclusively of middle-aged and older Chinese adults, our results may not fully reflect the association between ePWV and stroke risk in populations with different genetic backgrounds, lifestyles, or environmental exposures. For instance, genetic predispositions affecting vascular function, as well as differences in dietary patterns, air pollution exposure, and healthcare access across regions or ethnicities, may

modify the strength or direction of the observed associations. Further validation is needed in other racial and national populations. Second, the exclusion of individuals with missing data may introduce selection bias. Moreover, while we adjusted for known potential confounders, residual confounding factors that were not measured or controlled for may still exist. Lastly, due to the unavailability of stroke subtype information in CHARLS, we were unable to differentiate between ischemic and hemorrhagic stroke. Future studies should explore the relationship between ePWV and different stroke subtypes.

## 5 Conclusion

This study provides important evidence that elevated ePWV is significantly associated with increased stroke risk in middle-aged and elderly Chinese individuals. As a simple and non-invasive surrogate marker of arterial stiffness, ePWV may serve as a useful tool for early identification of individuals at high risk of stroke in clinical settings. Future research should investigate this association in more diverse populations and explore the relationship between ePWV and specific stroke subtypes, such as ischemic and hemorrhagic stroke, to enhance the clinical applicability of ePWV in stroke prevention strategies.

## Supporting information

**S1 File. Data.**
(XLSX)

## Acknowledgments

The authors thank all the members of the CHALRS for their contributions and the participants who contributed their data.

## Author contributions

**Conceptualization:** Xiaomei Zheng, Yixian Zhang.

**Data curation:** Shengren Xiong.

**Methodology:** Xiaomei Zheng, Yixian Zhang.

**Project administration:** Shengren Xiong.

**Software:** Pengsheng Chen.

**Supervision:** Shengren Xiong.

**Writing – original draft:** Xiaomei Zheng.

**Writing – review & editing:** Shengren Xiong.

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
