## [Decision Letter · Decision Letter 0]

9 Apr 2025

Dear Dr. Xiong,

Thank you for submitting your manuscript to PLOS ONE. After careful consideration, we feel that it has merit but does not fully meet PLOS ONE’s publication criteria as it currently stands. Therefore, we invite you to submit a revised version of the manuscript that addresses the points raised during the review process.

We look forward to receiving your revised manuscript.

Kind regards,

Sepiso K. Masenga, PhD

Academic Editor

PLOS ONE

Journal Requirements:

2. We note that there is identifying data in the Supporting Information file <S1 Dataset>. Due to the inclusion of these potentially identifying data, we have removed this file from your file inventory. Prior to sharing human research participant data, authors should consult with an ethics committee to ensure data are shared in accordance with participant consent and all applicable local laws.

-Location data

Please remove or anonymize all personal information (ID, AGE), ensure that the data shared are in accordance with participant consent, and re-upload a fully anonymized data set. Please note that spreadsheet columns with personal information must be removed and not hidden as all hidden columns will appear in the published file.

Reviewers' comments:

Reviewer's Responses to Questions

**Comments to the Author**

1. Is the manuscript technically sound, and do the data support the conclusions?

Reviewer #1: Yes

Reviewer #2: Yes

2. Has the statistical analysis been performed appropriately and rigorously?

Reviewer #1: Yes

Reviewer #2: Yes

3. Have the authors made all data underlying the findings in their manuscript fully available?

Reviewer #1: Yes

Reviewer #2: Yes

4. Is the manuscript presented in an intelligible fashion and written in standard English?

Reviewer #1: Yes

Reviewer #2: Yes

Reviewer #1: Clarity of Research Gap and Contribution

- The introduction effectively establishes the research gap by highlighting the limited studies on the relationship between ePWV and stroke risk, particularly in Asian populations. However, the novelty of the study could be further emphasised by contrasting it more explicitly with prior research. A more detailed comparison of their findings and methodologies would strengthen the justification for this study.

Statistical Analysis and Model Adjustments

- The statistical analysis is robust, with appropriate adjustments for demographic, socioeconomic, and cardiometabolic factors. However, the rationale for selecting specific covariates, such as smoking, alcohol consumption, BMI, etc., could be better explained. Are these factors known to confound the relationship between ePWV and stroke risk? A brief justification would enhance transparency.

- The use of restricted cubic spline analysis to explore the dose-response relationship is commendable. However, the choice of knots (10th, 50th, and 90th percentiles) should be justified. Were these percentiles chosen based on prior literature or exploratory analysis?

Methods

- The description of the CHARLS cohort is clear, but more details on the sample's representativeness would be helpful. For example, how does the demographic profile of the study participants compare to the general Chinese population?

Generalizability and Limitations

- The study acknowledges its limitations, including the lack of racial and regional diversity and potential selection bias due to missing data. However, the discussion could delve deeper into how these limitations might affect the interpretation of the results. For example, how might the findings differ in populations with different genetic or environmental risk factors?

- The exclusion of stroke subtypes (ischemic vs. hemorrhagic) is a significant limitation. The authors should elaborate on why this differentiation was not made and how it might impact the clinical applicability of the findings.

Clinical Implications

- The conclusion could be strengthened by briefly reiterating the clinical implications and suggesting directions for future research, such as investigating ePWV in diverse populations or exploring its relationship with stroke subtypes.

Reviewer #2: The authors have crafted a scientific and technically sound manuscript, and they have made a clear conclusion based on the data collected and analysed. All relevant data are within the paper and its Supporting Information files.

However, authors can correct one or two issues.

1. Under Table 1. Characteristics of study participants according to ePWV. Variables "rural" and "Illiteracy", the numbers are quite high 6141(66.82%) and 4358(47.42%) respectively. it would be insightful if authors can give a brief statement in relation to occurrence of stroke in the Chinese population.

2. Figure 1. Flowchart of study participant inclusion. Improve it by matching what has been written under the section "Study population and design." The sentence; "Had missing age information". If it can be added to the sentence that reads - "777 participants aged < 45" as it has been stated in text "we excluded individuals

who were under 45 years of age or had missing age information (n = 777)".

3. Authors mention that "Had incomplete follow-up data." but the flowcharts states that "3595 participants who were lost during follow up." If authors can relook at this and make it uniform for clarity.

4. Data Availability Statement

All relevant data arewithin the paper and its Supporting Information files. Add space between "are" and "within".

**Do you want your identity to be public for this peer review?** For information about this choice, including consent withdrawal, please see our Privacy Policy

Reviewer #1: **Yes: ** Situmbeko Liweleya

Reviewer #2: No

---

## [Decision Letter · Decision Letter 1]

22 Oct 2025

Association between estimated pulse wave velocity and stroke in middle-aged and elderly Chinese: a nationwide cohort study

PONE-D-25-06790R1

Dear Dr. Xiong,

We’re pleased to inform you that your manuscript has been judged scientifically suitable for publication and will be formally accepted for publication once it meets all outstanding technical requirements.

Kind regards,

Satish G Patil, PhD

Academic Editor

PLOS ONE

Additional Editor Comments (optional):

Reviewers' comments:

Reviewer's Responses to Questions

**Comments to the Author**

Reviewer #2: All comments have been addressed

2. Is the manuscript technically sound, and do the data support the conclusions?

Reviewer #2: Yes

3. Has the statistical analysis been performed appropriately and rigorously?

Reviewer #2: Yes

4. Have the authors made all data underlying the findings in their manuscript fully available?

Reviewer #2: Yes

5. Is the manuscript presented in an intelligible fashion and written in standard English?

Reviewer #2: Yes

Reviewer #2: All concerns raised in the last submitted manuscript have been addressed by the authors. I have no further concerns.

**Do you want your identity to be public for this peer review?** For information about this choice, including consent withdrawal, please see our Privacy Policy

Reviewer #2: No

---

## [Editor Report · Acceptance letter]

PONE-D-25-06790R1

PLOS ONE

Dear Dr. Xiong,

I'm pleased to inform you that your manuscript has been deemed suitable for publication in PLOS ONE. Congratulations! Your manuscript is now being handed over to our production team.

Kind regards,

on behalf of

Prof. Dr. Satish G Patil

Academic Editor

PLOS ONE